# Double-Stranded RNA Binding Proteins in Serum Contribute to Systemic RNAi Across Phyla—Towards Finding the Missing Link in Achelata

**DOI:** 10.3390/ijms21186967

**Published:** 2020-09-22

**Authors:** Thomas M. Banks, Tianfang Wang, Quinn P. Fitzgibbon, Gregory G. Smith, Tomer Ventura

**Affiliations:** 1GeneCology Research Centre, School of Science and Engineering, University of the Sunshine Coast (USC), 4 Locked Bag, Maroochydore, Queensland 4558, Australia; tmb042@student.usc.edu.au (T.M.B.); twang@usc.edu.au (T.W.); 2Institute for Marine and Antarctic Studies (IMAS), University of Tasmania, Private Bag 49, Hobart, Tasmania 7001, Australia; quinn.fitzgibbon@utas.edu.au (Q.P.F.); gregory.smith@utas.edu.au (G.G.S.)

**Keywords:** decapod crustaceans, dsRNA transport, gene silencing mechanism, serum dsRNA binding proteins, systemic RNAi

## Abstract

RNA interference (RNAi) has become a widely utilized method for studying gene function, yet despite this many of the mechanisms surrounding RNAi remain elusive. The core RNAi machinery is relatively well understood, however many of the systemic mechanisms, particularly double-stranded RNA (dsRNA) transport, are not. Here, we demonstrate that dsRNA binding proteins in the serum contribute to systemic RNAi and may be the limiting factor in RNAi capacity for species such as spiny lobsters, where gene silencing is not functional. Incubating sera from a variety of species across phyla with dsRNA led to a gel mobility shift in species in which systemic RNAi has been observed, with this response being absent in species in which systemic RNAi has never been observed. Proteomic analysis suggested lipoproteins may be responsible for this phenomenon and may transport dsRNA to spread the RNAi signal systemically. Following this, we identified the same gel shift in the slipper lobster *Thenus australiensis* and subsequently silenced the insulin androgenic gland hormone, marking the first time RNAi has been performed in any lobster species. These results pave the way for inducing RNAi in spiny lobsters and for a better understanding of the mechanisms of systemic RNAi in *Crustacea*, as well as across phyla.

## 1. Introduction

In 1998, Fire et al. demonstrated potent and efficient gene silencing in the nematode *Caenorhabditis elegans,* which built on the phenomenon of antisense RNA technology [1]. Injection of sense and antisense RNA for the myofilament protein coding gene *unc-22*, which formed double-stranded RNA (dsRNA), showed the twitching phenotype associated with *unc-22* knockdown [1]. Since then, the use of exogenous dsRNA to elicit gene silencing has become known as RNA interference (RNAi) and has developed into a widely used molecular tool to study gene function [2]. Apart from being a useful method to study gene function, RNAi also holds great potential in biotechnology, in areas such as pest management, molecular therapeutics, and aquaculture [2,3].

### 1.1. RNAi and Its Use in Decapods

The exogenous small interfering RNA pathway (exo-siRNA), which is associated with exogenous dsRNA-induced gene silencing, is a naturally occurring mechanism with roles in innate antiviral immunity and defence from mobile genetic elements [4] (Figure 1). Differences in the intracellular RNAi pathway are present in different lineages and taxa, although all begin with dsRNA first entering the cell and being recognized by the RNAse III protein, Dicer [4]. The L-shaped Dicer protein is able to bind and cleave dsRNA molecules into short 21–23 nucleotide RNA duplexes termed small interfering RNAs (siRNA), with different proteins such as R2D2 also involved, depending on the species [5,6]. These siRNA molecules are then loaded with Argonaute proteins and various cytoplasm proteins (such as component 3 promoter of RISC (C3PO), transactivation response element RNA-binding protein (TRBP), and others depending on species) to form the RNA-induced silencing complex (RISC), which unwinds the RNA duplex, discards or degrades the passenger strand, and seeks out the complementary mRNA molecule via Watson–Crick base pairing [7,8]. Once bound to a homologous mRNA molecule, Argonaute degrades it enzymatically, effectively silencing and knocking down the gene [9].

In decapod crustaceans, gene silencing is utilized broadly for research and aquaculture purposes. The main mode of gene silencing employed in decapod crustaceans is through exogenous dsRNA of 150–750 bp [3]. In the tiger prawn, *Penaeus monodon,* as well as the Australian redclaw crayfish, *Cherax quadricarinatus*, exogenous RNAi has been utilized multiple times successfully with high efficiency [10,11,12,13]. In two spiny lobster species, *Sagmariasus verreauxi* and *Panulirus ornatus*, gene silencing did not work successfully in multiple attempts (personal communication). This disconnect in silencing ability among decapod crustaceans limits the potential for RNAi-based research in spiny lobsters. The precise mechanisms of gene silencing in spiny lobsters remain elusive, and this study aims to illuminate the potential limiting factors in dsRNA-induced exogenous RNAi in these important aquaculture species, while providing insight into the extracellular components of this pathway.

### 1.2. Known Mechanisms of Systemic RNAi—A Drop in the Ocean

One of the more remarkable features of exogenous RNAi is its capacity to spread systemically across a wide variety of tissues and facilitate whole-animal gene knockdown [14]. Compared to the wealth of knowledge about the highly conserved core RNAi machinery, such as Dicer and Argonaute, little is known about systemic RNAi, which can vary drastically between taxa [14]. Systemic RNAi employs a partnership between dsRNA uptake and transport mechanisms, which in general are not well-characterized outside of model species, such as *Drosophila melanogaster* and *C. elegans* [14,15]. In terms of dsRNA uptake, two mechanisms are known to be involved—passive transport via dsRNA selective transmembrane channels, such as the well-studied SID-1, and receptor-mediated endocytosis [16,17]. SID-1 proteins first identified in *C. elegans* allow passive transport of dsRNA into the cell and are found ubiquitously in metazoan evolution, however whether they contribute to systemic RNAi seems to vary [18,19]. In crustaceans, however, SID-1 proteins in *Macrobrachium rosenbergii* and *Litopenaeus vannamei* were shown to facilitate dsRNA uptake [20,21,22]. Uptake of dsRNA may also occur via receptor-mediated endocytosis, as has been demonstrated in *C. elegans* and some insect species [17,23,24,25]. The *Caenorhabditis-*specific SID-2 receptor, scavenger receptors, and pattern recognition receptors have all been implicated in this mechanism [17,23,26]. Inhibiting clathrin with either pharmacological inhibitors or RNAi showed significantly decreased silencing efficiency in *D. melanogaster, Diabrotica vigifera vigifera*, *Tribolium castaneum*, and *C. elegans* [25,27,28]. As SID-1 proteins do not appear to be involved in systemic RNAi in insects, it is suggested dsRNA uptake via receptor-mediated endocytosis is the primary uptake mechanism [19,29]. This is exemplified in *T. castaneum*, where gene silencing is robust, SID-1 proteins definitely do not contribute to systemic RNAi, and blocking clathrin cripples the RNAi response [19,27,30]. In crustaceans, both SID-1 channels and receptor-mediated endocytosis appear to be involved in dsRNA uptake, although expression is tissue-dependent, suggesting tandem roles in systemic RNAi [21,31]. In *L. vannamei*, receptor-mediated endocytosis of dsRNA was only observed in the hepatopancreas, while SID-1 activity was observed in the muscles, gills, and hemocytes, but not in the hepatopancreas [21,31]. This potentially represents a mechanism in which dsRNA delivered orally is endocytosed into hepatopancreas cells via an unknown receptor and then spread to other tissues via SID-1 proteins to elicit systemic RNAi, which has been observed via oral delivery previously [32].

### 1.3. dsRNA Transport in Serum

Transport of the RNAi signal throughout the body is a poorly understood, yet seemingly vital mechanism of systemic RNAi in invertebrate species with more complex circulatory systems, such as arthropods. To date, only a handful of proteins involved in this transport mechanism have been identified. In 2009, Sakishita et al. showed that Apolipophorin 1 (Apo-1) was able to bind dsRNA in the hemolymph of *Bombyx mori* [33]. When *B. mori* serum was incubated with dsRNA, dsRNA migration was inhibited significantly on an agarose gel, with comparable results being achieved later in *Blattodea*, *Diptera*, and *Orthoptera* [33,34]. Lipophorins are known to be involved in innate immunity, and given that dsRNA is a known viral pattern associated molecular pattern (PAMP), this phenomenon may be part of the antiviral defence mechanism in invertebrates [35]. Furthermore, this hypothesis is strengthened by the fact that lipophorin is the ligand for scavenger receptors, which facilitate systemic RNAi in insects [17,35]. The only other identified serum dsRNA binding protein was the major royal jelly protein 3 (MJRP3) in *Apis melifera* [36,37]. As this protein is secreted in the royal jelly, it is believed to be involved in social immunity and transmits dsRNA horizontally, as opposed to lipophorin, which is linked to innate immunity [36,37]. No such gel shift has been observed in any crustacean species previously, nor have any dsRNA binding proteins been identified in the hemolymph. The only other instances of dsRNA transport in invertebrate species are via exosomes in *T. castaneum* cell lines and hemocytes in vivo in *D. melanogaster* [38,39].

We show here a cross-phyla gel trend, where the presence of dsRNA binding proteins in the serum of various invertebrate and vertebrate species is correlated with successful induction of exogenous RNAi utilizing dsRNA. Following this, the putative dsRNA binding proteins responsible are identified, and using this knowledge exogenous RNAi is attempted and achieved in the slipper lobster, *Thenus australiensis*, which marks the first instance of gene silencing being induced in any lobster species. These findings pave the way for inducing the comparatively elusive gene silencing in spiny lobster species and for a better understanding of the systemic RNAi mechanism across phyla.

## 2. Results

### 2.1. Correlation between Humoral Response to dsRNA and Silencing Capacity Using Exogenous dsRNA Across Phyla

An electromobility shift assay (EMSA) revealed that 1 µg of dsRNA pre-incubated with 10 µL of serum from prawn species *P. monodon*, *M. rosenbergii*, and crayfish species *C. quadricarinatus* migrated slower than non-incubated dsRNA, with a retarded band of slower mobility than the 8 kb DNA marker, compared to the 660 bp band of the lone dsRNA (Figure 2). The serum from three spiny lobster species, *S. verreauxi*, *P. ornatus*, and *Panulirus homarus*, meanwhile did not affect the migration of dsRNA (Figure 2). Interestingly, this gel shift correlates with in vivo dsRNA-induced RNAi, which has been observed in *C. quadricarinatus*, *P. monodon*, and *M. rosenbergii*, but not in any spiny lobster species to date [10,12,40]. A ten minute incubation of dsRNA and serum was enough to elicit this gel shift—the results did not change after incubation for a week at room temperature (~21–23 °C). Pre-incubating the serum of *C. quadricarinatus* with proteinase K at 37 °C overnight prior to incubation with dsRNA eliminated the dsRNA gel shift, indicating that the binding is due to protein(/s). A similar result was obtained when pre-heating the serum at 95 °C for five minutes, indicating that the gel shift is due to heat sensitive protein(/s) (Figure 3A). Diluting the serum in a series of ratios of 1:2, 1:4, 1:8, 1:16, 1:32, and 1:64 revealed that the minimum hemolymph concentration that is required to elicit this gel shift was one part serum for eight parts water (1:8) (Figure 3B). This correlation between exogenous RNAi and the observed gel shift was further explored with a variety of species across phyla.

When dsRNA was incubated with serum from three fish species, *Oreochromis mosambicus*, *Seriola lalandii*, and *Epinephelus lanceolatus*, only the latter inhibited dsRNA migration, comparable to what was observed earlier in crayfish and prawns (Figure 4A). The correlation between this gel shift and gene silencing is again seen as an exogenous RNAi-based vaccine against the nervous necrosis virus for *E. lanceolatus* that has been observed [41]. Exogenous RNAi has been reported in humans, so the dsRNA binding capacity of human serum was tested as well, yielding a result comparable to crayfish and prawns [42] (Figure 4A). An inhibited band with a mobility slower than 10 kbp was observed when dsRNA was incubated with human serum, while non-incubated dsRNA had a band of 660 bp. It is worth noting, however, that long dsRNA molecules trigger the interferon response in mammalians, so any application of exogenous RNAi in humans was done with siRNA [43].

The sera of two echinoderms were also tested for dsRNA binding activity. The migration of dsRNA was affected very marginally when incubated with *Acanthaster planci* and *Holothuria leucospilota* sera, with a retarded band migrating slower than ~700–750 bp, as opposed to the non-incubated dsRNA at 660 bp (Figure 4A). Exogenous RNAi has been observed in *H. leucospilota*, which is consistent with the trend we have observed; however, the gel shift is not as pronounced as that observed in prawns and crayfish. No report of exogenous dsRNA-induced RNAi, however, has been observed in *A. planci*. 

Incubation of dsRNA with sera from various mollusk species, including *Saccostrea glomerata*, *Crassostrea gigas*, and *Aplysia dactylomela* (Figure 4A,C) showed a modest inhibition of dsRNA migration with a retarded band with a mobility of ~1 kbp in size, while the lone dsRNA remained at 660 bp. This gel shift appears to be unique to *Mollusca*. In *C. gigas*, exogenous RNAi has been observed, which is consistent with our trend, while exogenous RNAi has not been reported in the other mollusks we tested [44]. 

The serum from *B. mori* larva when incubated with dsRNA yielded a result comparable to that obtained by Sakashita et al. (2009) [33]. When incubated with *B. mori* serum, the dsRNA band was >4 kbp, while the dsRNA without serum was at 660 bp. Exogenous RNAi has been demonstrated in *B. mori*, which is consistent with our trend [45].

### 2.2. Identification of Putative dsRNA Binding Protein Complex in Sera via Mass Spectrometry Analysis

Analysis of the dsRNA–protein complex in *C. quadricarinatus*, *E. lanceolatus*, and *B. mori* revealed a list of immune-, lipid-transport-, oxygen-transport-, and coagulation-related proteins, which was to be expected of serum samples (Table 1). In *C. quadricarinatus*, hemocyanin was identified, as well as proteins involved with mRNA methylation, translation initiation, calcium transport, and protein acetylation. The most likely candidate for dsRNA binding in serum was a 1071 amino acid protein identified as a clotting protein precursor via NCBI BLAST, containing a Vitellogenin N, d1lsha3, coiled coil, and von Willebrand factor type D (VWD) domains (Figure 5A).

In *E. lanceolatus*, far more proteins were identified, which was to be expected given the relative complexity of vertebrate species. Hemoglobin was identified, along with various complement proteins, immunoglobulins, prothrombin, fibrinogen, and ceruloplasmin. Apolipoproteins were also identified and are most likely the dsRNA binding proteins given their previous implication in dsRNA and siRNA binding (Figure 5B) [34,46].

In *B. mori* only lipoproteins and hemocyanin were identified in the dsRNA–protein complex. It has been shown previously that Apolipophorin-1 is the causative agent responsible for this gel shift and is able to bind dsRNA in the hemolymph of various insect species, including *B. mori* (Figure 5C) [33,35]. Interestingly, only Apolipophorin-3 was identified in our mass spectrometry results, not Apolipophorin-1. For a full list of identified proteins, see Appendix A.

### 2.3. Following the Gel Shift in Crustaceans 

In *C. quadricarinatus*, the proposed dsRNA binding protein was a clotting factor, so different anticoagulants were tested to observe any changes in the gel shift. When using a TBE buffer as an anticoagulant, the *C. quadricarinatus* hemolymph did not coagulate and the gel shift was absent when incubating with dsRNA (Figure 6). The hemolymph with an anticoagulant optimized for freshwater species did not coagulate, and showed a result comparable to our original result when pre-incubated with dsRNA, with an inhibited band with a mobility slower than 10 kb, as opposed to 660 bp for lone dsRNA (Figure 6). The hemolymph mixed with an anticoagulant optimized for marine species partially clotted, and when incubated with dsRNA, showed an intermediary result between the TBE and freshwater anticoagulant, with a large smear at 500–1000 bp as opposed to 660 bp for non-incubated dsRNA (Figure 6).

Following on from the across-phyla gel shift trend result, this trend was studied further in two crustacean species, *Cherax destructor* and *Thenus australiensis*. In *C. destructor*, a gel shift comparable to that seen in *C. quadricarinatus* was observed when incubating serum with dsRNA. *C. destructor* serum + dsRNA showed a band of retarded mobility >3 kb compared to non-incubated dsRNA at 660 bp (Figure 4D). There have been no reports of exogenous RNAi in *C. destructor*, although it is plausible to hypothesize that it may be functional, given how closely related it is to *C. quadricarinatus,* where exogenous RNAi has been applied many times [3,47].

When incubating *T. australiensis* serum with dsRNA, a gel shift was observed similar to those in crayfish and prawns (Figure 4B). No anticoagulant was required, and dsRNA incubated with *T. australiensis* serum migrated slower than unincubated dsRNA, with bands at >4 kb and 660 bp, respectively. This is interesting, as when testing for dsRNA binding in other members of the Achelata taxa (i.e., spiny lobsters), this gel shift was not observed. Given our findings that this gel shift correlates with the capacity of dsRNA-induced silencing, it was plausible that exogenous RNAi could be performed in *T. australiensis*.

### 2.4. dsRNA-Induced IAG Knockdown in T. australiensis

Our qPCR assay revealed highly significant reduction of *Thenus australiensis* insulin androgenic gland hormone (*Taus-IAG)* expression relative to 18s in the animals injected with *Taus-IAG* dsRNA compared to the group injected with GFP dsRNA, with an estimated silencing efficiency of 95% (Figure 7). Here, 5 µg/g of dsRNA per animal was enough to elicit *Taus-IAG* knockdown. This marks the first time dsRNA gene silencing has been induced in any lobster species.

## 3. Discussion

The precise mechanisms of systemic RNAi appear to differ greatly across lineages, yet here we describe a trend across phyla relating to dsRNA transport and stability in vivo. The trend of dsRNA binding activity of the serum appears to be consistent with the exogenous RNAi capacity (with the exception of species in which silencing has not been tested, such as *S. lalandii*, *A. dactylomela*, *S. glomerata*, and *A. planci*). These results indicate that dsRNA binding proteins in the serum contribute to the RNAi response in a variety of species and may be limiting factors in species in which RNAi is seemingly unfunctional, such as spiny lobsters (personal communication). It is likely that these dsRNA binding proteins protect and shuttle dsRNA in the serum to be delivered to cells where uptake occurs, either via dsRNA channels such as SID-1 or receptor-mediated endocytosis, which is consistent with findings from Sakashita et al. (2009), Wynant et al. (2014), and Maori et al. (2019) [17,33,35,36,37]. The presence of these serum dsRNA binding proteins, however, only seems to indicate the capacity to silence genes, not necessarily high silencing efficiency. When testing *B. mori* serum, dsRNA binding activity was observed, as seen previously, yet the species is known to be highly recalcitrant to RNAi, similarly to other lepidopterans, which suggests these dsRNA transport proteins enable the capacity for systemic RNAi but do not necessarily enhance the response [48]. As mentioned, a few of the species we tested lack information on whether exogenous RNAi works, however given the observed trend some hypotheses can be drawn. No gel shift was seen when incubating *O. mosambicus* serum with dsRNA (Figure 4A), however there is one report of RNAi working in vitro in a closely related tilapia species, *Oreochromis niloticus* [49]. Knockdown of the tilapia *gas2* gene was facilitated by transfecting tilapia brain cells with short hairpin loop RNA containing plasmids with reasonable success [49]. These species are closely related, which would suggest that RNAi may work in *O. mosambicus*, however our trend correlates in vivo exogenous RNAi and dsRNA binding of the serum, not in vitro RNAi. The lack of dsRNA binding proteins in the *O. mosambicus* serum may prevent in vivo exogenous RNAi from functioning, however in a cell culture system where dsRNA transport is not a limiting factor, RNAi is applicable [50]. This may also be the case in *S. lalandii*, where an identical gel response was observed (Figure 4A). The sera of *A. planci* and *H. leucospilota* + dsRNA led to a very marginal but still apparent gel shift (Figure 4A), and exogenous RNAi works in the latter, which suggests capacity for RNAi in the former [51]. As the gel shift for these echinoderm species was difficult to determine, the protocol was repeated and a clearer gel shift was observed akin to that seen in *C. gigas* (S1). In *C. gigas*, where silencing works efficiently, dsRNA + serum elicited a modest gel shift, with identical results in *S. glomerata* and *A. dactylomela* [44]. Exogenous RNAi may work in *S. glomerata* as they belong to the same family, although it is too farfetched to claim the same for *A. dactylomela*, as they are a different order of mollusk, and thus more research is needed to confirm whether silencing works in these species.

Previously, lipoproteins and other lipid binding proteins have been implicated in nucleic acid binding and innate immunity, which is consistent with our results implicating them in dsRNA binding [33,34,35,52,53] (Table 1 and Figure 5). The dsRNA binding protein candidates identified in *C. quadricarinatus*, *E. lanceolatus*, and *B. mori* via mass spectrometry all share conserved functions in terms of lipid binding capacity, evidence of dsRNA binding, and links to innate immunity, but differ greatly in structure and size. In *C. quadricarinatus*, the putative dsRNA binding protein in the serum was a clotting factor, with a coiled coil domain indicating the potential to bind dsRNA and other nucleic acids [54]. This protein also contains a VWD domain at the C-terminus; such domains are known to be involved in multimerization (Figure 5) [55]. When suboptimal anticoagulants were used, the gel shift was altered in *C. quadricarinatus*, suggesting that clotting factors are at least partially involved in the formation of the dsRNA–protein complex (Figure 6). This clotting protein may bind dsRNA with the coiled coil structure, and VWD domains may link to form large complexes that deliver the RNAi signal to cells via an unknown uptake mechanism. This is plausible given that the exo-siRNA pathway is antiviral in nature and clotting proteins represent a major part of the crustacean immune response [56]. In *E. lanceolatus*, apolipoprotein A-1 was identified as the putative dsRNA binding protein. Apolipoprotein A-1 is known to possess antiviral activity and bind siRNA, while also being a prevalent serum protein [34,52]. These characteristics make it the most likely dsRNA binding protein identified in *E. lanceolatus* serum. *B. mori* Apolipophorin-1 is the dsRNA binding protein responsible for the gel shift, as reported previously [33]. Domain analysis of the protein via SMART revealed a C-terminus structure similar to that observed in the clotting protein, with putative dsRNA binding capability in *C. quadricarinatus*, with a coiled coil and VWD domain. This may operate on a similar mechanism to the *C. quadricarinatus* protein, with Apolipophorin-1 binding dsRNA via coiled coils and forming large multimers with the VWD domain before being endocytosed by scavenger receptors [53]. Interestingly, we did not identify Apolipophorin-1 in our *B. mori* serum samples, whereas others did in *B. mori* and other insect species [33,35]. This may be due to a difference in methodology, as Sakashita et al. (2009) used chromatography to isolate dsRNA binding proteins, Wynant et al. (2014) used centrifugation to extract the gel shift band, while we used further electrophoresis (Figure 8) [33,35]. Both also used Edman degradation to identify Apolipophorin-1, while we used mass spectrometry utilizing publicly available ORF datasets [33,35]. Overall, proteins involved in lipid transport and binding initially do not seem to be prime dsRNA binding candidates given the significant structural differences between lipids and nucleic acids. It can be speculated, however, that this dsRNA binding function may have evolved into lipid binding proteins due primarily to their prevalence in the serum. The serum has access to many if not all tissues, and the abundance of lipoproteins would in turn allow for large amounts of dsRNA to be transported to sites of viral infection. The mechanisms of lipid uptake into cells may also have been co-opted during evolution to facilitate dsRNA uptake as well. This is supported by studies in locusts, where Apolipophorin binds both dsRNA and lipids and is a ligand for scavenger receptors, which have been implicated in dsRNA uptake [17,35]. More research is needed, however, to validate this speculation in a broader cross-phyla regard.

We identified a gel shift when incubating the serum from *C. destructor* with dsRNA comparable to that in *C. quadricarinatus*, suggesting it is quite likely silencing can be achieved in this species, given its close relation to *C. quadricarinatus*, where exogenous RNAi is inducible with high efficiency and consistency [10,11]. It is reasonable to assume the dsRNA binding protein responsible for the *C. destructor* gel shift is homologous to that in *C. quadricarinatus* and possesses a similar mechanism. The strongest evidence for these dsRNA binding proteins facilitating RNAi comes from our spiny and slipper lobster results (Figure 4B and Figure 7). The capacity for exogenous RNAi exists in many decapod lineages, while all attempts to silence genes in spiny lobsters have resulted in failure [10,11,12,13,40]. Considering that exogenous RNAi has been reported in a variety of non-decapod crustaceans as well, such as *Daphnia magna* and *Artemia franciscana*, it is possible that the extracellular components necessary to elicit spiny lobster systemic RNAi have been lost during evolution [3]. The exo-siRNA pathway is antiviral in nature and represents a major pathogen defense mechanism in invertebrates—if this has been hindered in spiny lobster species, they should be highly susceptible to viral infection. Interestingly, however, only a single virus has ever been isolated from any spiny lobster, which suggests a powerful antiviral immune system [57,58]. The contradiction of powerful antiviral mechanisms and a lack of an RNAi response indicates that other mechanisms that deal with viruses must exist within spiny lobsters. This is supported by the fact that extracting RNA is notoriously difficult in spiny lobsters due to rapid degradation, requiring powerful chemical agents such as beta-mercaptoethanol to denature their potent RNAses [59]. A mechanism surrounding these powerful RNAses may be responsible for the antiviral immune response in species such as *P. ornatus and S. verreauxi*, leading to a “vestigial” RNAi system. If this is the case, the need for dsRNA transport in the hemolymph is absent, which may explain the lack of a gel shift (Figure 3). This becomes more apparent when compared to the closely related slipper lobster, *T. australiensis*. In *T. australiensis*, the dsRNA+ serum gel shift is present, RNA does not degrade at all, and remarkably dsRNA-induced gene silencing is functional with high efficiency (Figure 4B and Figure 7). This difference in silencing capacity likely does not come from the lack of core RNAi machinery, such as Dicer and Argonaute, as they possess integral roles outside of RNAi, such as epigenetic modification and small RNA biogenesis; as such, it stands to reason they should be present in all metazoans [60,61]. Knowing this, the most probable limiting factors in exogenous RNAi for spiny lobsters are dsRNA uptake and more relevant dsRNA transport, which fits the trend we have observed. There is one major inconsistency in this conclusion, however, which is that spiny lobsters also possess clotting factors, which make up a major immune response and possess lipid transport proteins [56,62,63]. There must, therefore, be a “missing link” that is allowing these proteins to form complexes with dsRNA and deliver the dsRNA systemically, which is absent in spiny lobster species. The fact that silencing works in *T. australiensis* implies the presence of this “missing link” and prompts further comparative research between these species and across Decapoda to elucidate this seemingly integral factor in systemic RNAi.

## 4. Materials and Methods

### 4.1. Hemolymph, Hemocoel, Plasma, and Serum Collection

Hemolymph, hemocoel, plasma, and serum samples from a wide variety of species across phyla were used in this study, each with different methodologies for collection. All non-arthropod sera, with the except of *C. gigas* hemolymph, were donated by colleagues working on a variety of invertebrate and vertebrate species, with appropriate anticoagulants used where necessary. For crayfish and freshwater prawn hemolymph extraction, a 1 mL insulin syringe pre-coated with pre-cooled freshwater anticoagulant (140 mM NaCl, 10 mM KCl, 10 mM HEPES, 10 mM EDTA-Na_2_, 30 mM trisodium citrate, pH 7.3) was used. Hemolymph samples were drawn from the ventral abdominal sinus and mixed on ice with an equal volume of anticoagulant in a 1:1 ratio. For spiny lobsters, a pre-cooled marine anticoagulant (0.45 M NaCl, 0.1 M glucose, 30 mM trisodium citrate, 26 mM citric acid, 10 mM EDTA; pH 4.6) was used to pre-coat a 1mL insulin syringe, the hemolymph was drawn from the 5th walking leg sinus, then mixed in a 1:1 volume with anticoagulant on ice. *T. australiensis* hemolymph was drawn using the same method, however an anticoagulant was not needed. For marine prawns (*P. monodon* and *M. rosenbergii*), the hemolymph was drawn from the ventral abdominal sinus and the same anticoagulant was used as for spiny lobsters. For *B. mori*, the hemolymph was drawn from the dorsal blood vessel and stored on ice before freezing at −20 °C. For *C. gigas*, the hemolymph was drawn from the adductor muscle and chilled on ice before freezing at −20 °C.

### 4.2. Double-Stranded RNA Production

A total of 5 mg double-stranded RNA of *P. monodon* gonad inhibiting hormone (*dsPm-GIH*) was produced for this trial. RNA was extracted from female *P. monodon* central nervous system tissue using Trizol^®^ Reagent (Invitrogen, Carlsbad, California, United States), following the manufacturer’s instructions. RNA quality and quantity were assessed using a NanoDrop™ 2000 Spectrophotometer (Thermo Scientific, Waltham, Massachusetts, United States) and Agilent 2100 Bioanalyzer (Agilent Technologies, Inc, Santa Clara, California, United States), followed by reverse transcription of 1 ug RNA into cDNA using a Tetro cDNA synthesis kit (Bioline, London, United Kingdom), as per the manufacturer’s instructions. *Pm-GIH* was then amplified from this cDNA sample with primers (Pm-GIH_F: 5′-TGCAAGCAATGAAAACATGG-3′ and Pm-GIH-R: 5′-CCAATCAGTTCCCCTTGAAA-3′) derived from publicly available sources (Pm-GIH, GenBank Accession number: DQ643389.1). The resultant 660 bp amplicon was then visualized with gel electrophoresis (1.2% agarose) and purified with the QIAquick PCR Product Purification Kit (Qiagen, Melbourne, Australia), as per manufacturer’s instructions. The quality was assessed via nanodrop, and amplicon was diluted ar 1:10,000 to serve as a template for on-top PCR. The on-top PCR included two sets of primers, where the T7 promoter sequence and six additional residues upstream of this sequence (5′–nnnnnnTAATACGACTCACTATAGGG–3′; T7P) were extended onto the original primers. For the PCR reactions, one set of primers included the T7P attached to Pm-GIH_F and an intact Pm-GIH_R, while the other set of primers included the intact Pm-GIH_F and T7P attached to the Pm-GIH_R. Amplicons were visualized with gel electrophoresis (1.2% agarose) and then purified using a QIAquick PCR Product Purification Kit (Qiagen, Melbourne, Australia), as per the manufacturer’s instructions. The cleaned amplicons with T7P flanking from either the 5′ or the 3′ of the sequence served as templates for in vitro RNA transcription using the HiScribe T7 High-Yield RNA Synthesis Kit (New England Biolabs, Melbourne, Australia), as per the manufacturer’s protocol. Following overnight incubation at 37 °C, the RNA was treated with DNase for 1 h at 37 °C, followed by cleaning with both chloroform-saturated acidic phenol and chloroform, then ethanol precipitation. To form *dsPm-GIH*, the two strands of RNA were mixed and joined via a 10 min incubation at 70 °C, followed by overnight incubation at room temperature. The stability and purity of the *dsPm-GIH* were assessed using gel electrophoresis on a 1.2% agarose.

### 4.3. Electromobility Shift Assay (EMSA)

Here, 1 ug *Pmon-GIH* dsRNA was incubated with 10 ul hemolymph. which was pre-diluted 1:1 with anticoagulating solution. Hemolymph samples were centrifuged for five minutes at 5000× *g* to remove cell debris and hemocytes. The initial incubation period was 10 min at room temperature, then the incubation was extended up to 7 days, with no additional effect on the result. Multiple hemolymph samples were incubated with multiple dsRNA from different species, with no change in result. We, therefore, proceeded with all experiments with the 10 min incubation up to 2 h, using the *Pmon-GIH* dsRNA. Following incubation, loading dye was added to the dsRNA–hemolymph mix, which was run on a 1.2% agarose gel (100 V, 50 min) and visualized under UV light using ethidium bromide.

### 4.4. uHPLC Tandom QTof Mass Spectrometry Analysis

Hemolymph samples from *C. quadricarinatus*, *Epinephelus lanceolatus,* and *B. mori* were incubated with dsRNA followed by EMSA as described above. Instead of 10 µL serum, 200 µL was used with 20 µg of *Pmon-GIH* dsRNA and 0.8% agarose. The bands were cut from the gel under UV light with a sharp scalpel, minced, then placed in a small cup fitted with a 3 kDa cutoff filter at its bottom to collect dsRNA and any protein bound to it. The cup was laid horizontally on the gel running apparatus and subjected to 100 V for another 50 min. The gel pieces were then removed from the cup and the dsRNA–protein solution was retrieved for protein purification. To validate the presence of the dsRNA in the solution, the solution, filter, and gel pieces were all visualized under UV light using ethidium bromide. The nucleic acid content and protein content of the solution were then subjected to quantification via nanodrop method. The dsRNA–protein solution was then lyophilized overnight and subjected to in-solution digestion as described elsewhere [64] (Figure 8). Briefly, about 500 μg total protein, estimated at 280 nm on a NanoDrop 2000 (Thermo Fisher Scientific, Bremen, Germany), was dissolved in 100 μL lysis buffer (8 M urea, 0.8 M NH_4_HCO_3_, pH 8.0), then reduced with 100 mM DTT at 37 °C, and subsequently alkylated with 100 mM IAA at room temperature (RT) in the dark, followed by the incubation with the addition of 100 mM DTT at RT. The urea concentration was reduced by diluting the mixture with MilliQ water, then the proteins were digested with 10 μg sequencing-grade modified trypsin at 37 °C overnight. The digested samples were desalted on Sep-Pak C18 columns (Waters, Milford, MA) and lyophilized for LCMS analysis.

The tryptic peptides were resuspended in 25 μL of 0.1% formic acid in MilliQ water and analyzed by a QTOF X500R mass spectrometer (AB SCIEX, Concord, Canada) equipped with an electrospray ion source attached to an ExionLC liquid chromatography system (AB SCIEX, Concord, Canada). The LCMS method was described in a previous study [65]. In brief, 20 μL of each sample was injected onto an Aeris™ 1.7 μm PEPTIDE XB-C18 100 Å uHPLC column (Phenomenex, Sydney, Australia) equipped with a SecurityGuard column. For LC fractionation, solvent A consisted of 0.1% (v/v) formic acid and solvent B contained 100% acetonitrile/0.1% formic acid. Linear gradients of 5–35% solvent B over 10 min at 400 μL/min flow rate, followed by a gradient from 35 to 80% solvent B in 2 min and 80 to 95% solvent B in 1 min, were used for peptide elution. Solvent B was held at 95% for 1 min to wash the column and returned to 5% solvent B for the next injection. The ion spray voltage was set to 5500 V, the declustering potential was set to 100 V, the curtain gas flow was set to 30, ion source gas 1 was set to 40, ion source gas 2 (GS2) was set to 50, and the spray temperature was set to 450 °C. The mass spectrometer acquired mass spectral data in an information-dependent acquisition mode. Full-scan TOF-MS data were acquired over the mass range of 350–1400, while for product ion the data were acquired over the MS/MS range of 50–1800. Ions observed in the TOF-MS scan exceeding a threshold of 100 cps and a charge state of +2 to +5 were set to trigger the acquisition of the product ion.

The LC-MS/MS data were converted using msconvert [66] and imported to the PEAKS studio (Bioinformatics Solutions Inc., Waterloo, ON, Canada, version 7.0). The *C. quadricarinatus* database included a transdecoder peptide file based on the transcriptome from publicly available databases (NCBI BioProject accession number PRJEB5112), as well as ORFs from our eyestalk database [67,68]. The *E. lanceola* database included a transdecoder peptide file based on the whole larvae transcriptome. The database for *B. mori* was retrieved from publicly available sources (retrieved: http://sgp.dna.affrc.go.jp/ComprehensiveGeneSet/). De novo sequencing of peptides, database searches, and characterization of specific post translational modifications were used to analyze the raw data; the false discovery rate (FDR) was set to ≤1%, and [−10 × log(P)] was calculated accordingly, where P is the probability that an observed match is a random event. The PEAKS software used the following parameters: (i) precursor ion mass tolerance, 0.1 Da; (ii) fragment ion mass tolerance, 0.1 Da (the error tolerance); (iii) tryptic enzyme specificity with two missed cleavages allowed; (iv) monoisotopic precursor mass and fragment ion mass; (v) a fixed modification of cysteine carbamidomethylation; (vi) variable modifications, including lysine acetylation, deamidation on asparagine and glutamine, oxidation of methionine, and conversion of glutamic acid and glutamine to pyroglutamate.

### 4.5. In Vivo T. Australiensis IAG Silencing

Twelve male juvenile slipper lobsters (size range: 25.7–84.3 g (mean 44.6/SD 17.2)) were split into two groups and injected with 5 µg/g of *Taus-IAG* dsRNA or GFP dsRNA. The dsRNA was purchased from Genolution (http://genolution.cafe24.com/default/index.php) and the sequence for *Taus-IAG* was retrieved from our 5th walking leg RNA-seq library (S2). Twenty-four hours post injection, animals were culled on ice and the 5th walking legs were dissected, snap frozen with liquid N_2_, and stored in −80 °C.

### 4.6. RNA Extraction and RT qPCR

RNA was extracted from the 5th walking leg (which contained the androgenic gland) using the methodology described earlier by Hyde et al. (2019) [59]. Briefly, the 5th walking leg tissue was mechanically homogenized in RNAzol^®^ RT reagent from the Molecular Research Center (Cincinnati, Ohio, United States) and 1% β–mercaptoethanol to prevent RNA degradation. Following the manufacturer’s protocol, RNA was extracted and the total RNA content and quality were quantified via nanodrop. RNA samples were then reverse-transcribed into cDNA and *Taus-IAG* expression was quantified via qPCR using the following primers: actcctctgagccccatttt (qTaus-IAG_F) and gcaagtcgagcaggtttctt (qTaus-IAG_R). *T. australiensis* 18s was retrieved from our transcriptome libraries and was used to normalize expression, with the following primers: ggtgcatggccgttctta (qTaus-18s_F) and tggagatccgtcgactagttaat (qTaus-18s_R). Reactions were carried out in the Rotor Gene 6000 thermocycler. The relative gene expression compared to 18s was then calculated using the 2^−ΔΔCT^ method.

### 4.7. Bioinformatic Analysis

Proteins sequences identified with the mass spectrometer were extracted in FASTA format and the best identity was determined by NCBI BLASTp, with an e-value cut off of 1 × 10^−30^. The domain architecture was then predicted using SMART and potential dsRNA binding proteins were noted [69].

## 5. Conclusions

In this study we have demonstrated the involvement of extracellular dsRNA binding proteins in systemic RNAi across phyla, suggesting dsRNA transport in the serum holds an integral role in the gene silencing mechanism. Incubation of serum with dsRNA inhibited dsRNA migration in gel electrophoresis, suggesting the presence of dsRNA binding proteins, which correlated with RNAi capacity. The most striking example of this was seen in our decapod models, in which dsRNA binding activity of the hemolymph was tested in numerous species. In almost all decapod species tested, a strong inhibition of dsRNA migration was observed, suggesting potent dsRNA binding proteins. A notable exception was the spiny lobster species, where no dsRNA binding was seen, and consequently RNAi has never been induced prior. With this knowledge, a gel shift was observed in the closely related slipper lobster *Thenus australiensis*, and subsequently IAG silencing was achieved via dsRNA injection. This disparity in silencing capacity between two Achelata species opens up an avenue for further research to discover and incorporate the missing link and limiting factor in spiny lobster RNAi, and induce gene silencing in this economically important species while understanding the systemic RNAi mechanisms in decapod crustaceans.

## Figures and Tables

**Figure 1 ijms-21-06967-f001:**
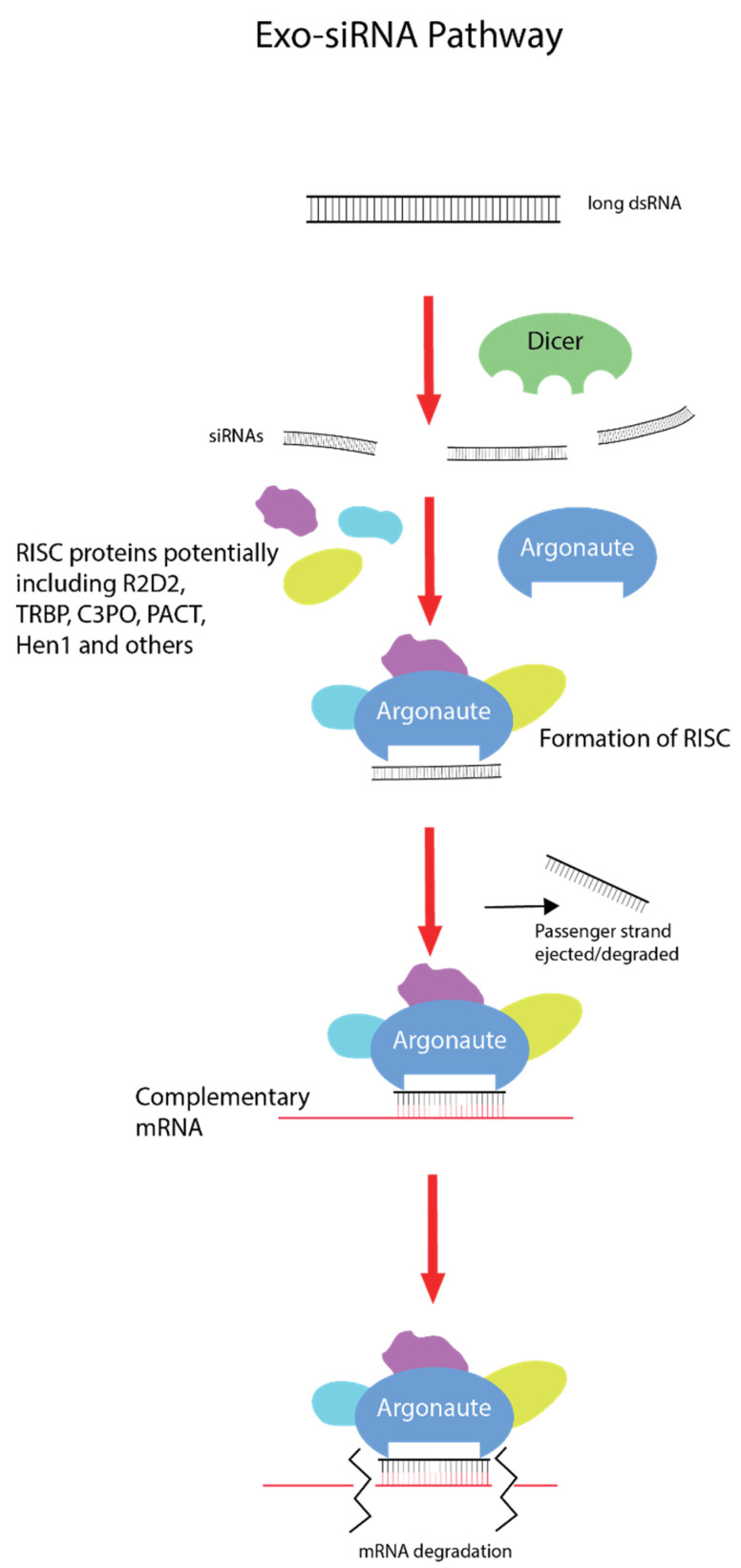
Exogenous small interfering RNA pathway (exo-SiRNA). Double-stranded RNA (dsRNA) enters the cell via different uptake mechanisms depending on the taxa. Long dsRNA molecules are bound by Dicer proteins, which cleave them into small 21–23 nucleotide double-stranded RNA duplexes called small interfering RNAs (siRNA). With the assistance of partner proteins such as R2D2 (varying across lineages), these siRNA molecules are loaded with Argonaute proteins and others (again dependent on lineage) to form the RNA-induced silencing complex (RISC). The RISC will then unwind and discard the passenger RNA strand and survey the cytoplasm for the homologous mRNA molecule. Once this mRNA is found, the guide RNA strand binds via Watson–Crick base pairing and Argonaute degrades the transcript, effectively silencing the gene.

**Figure 2 ijms-21-06967-f002:**
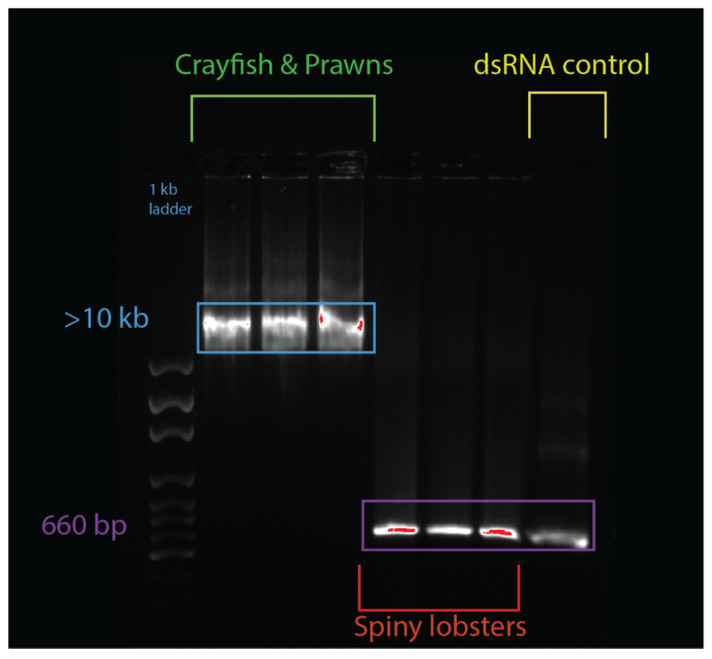
Serum + double-stranded RNA (dsRNA) gel shift observed in various decapod crustaceans. The lanes from left to right contain: 1 kbp ladder, *Cherax quadricarinatus* serum + dsRNA, Penaeus monodon serum + dsRNA, *Macrobrachium rosenbergii* serum + dsRNA, *Sagmariasus verreauxi* serum + dsRNA, *Panulirus ornatus* serum + dsRNA, *Panulirus homarus* serum + dsRNA, and a dsRNA + water control. The hemolymph from each species was spun down, then serum was collected and incubated with the same dsRNA for ten minutes. Following this incubation, samples were run on an agarose gel, and a mobility shift was observed in the crayfish and prawn species (in lanes 2–4), which was absent in spiny lobster species (lanes 5–7). The dsRNA control was 660 bp, while the dsRNA incubated with crayfish and prawn serum was at >10 kbp, indicating a protein–RNA interaction distinctly absent from the dsRNA incubated with the spiny lobster serum, which migrated normally to 660 bp, as seen in the control.

**Figure 3 ijms-21-06967-f003:**
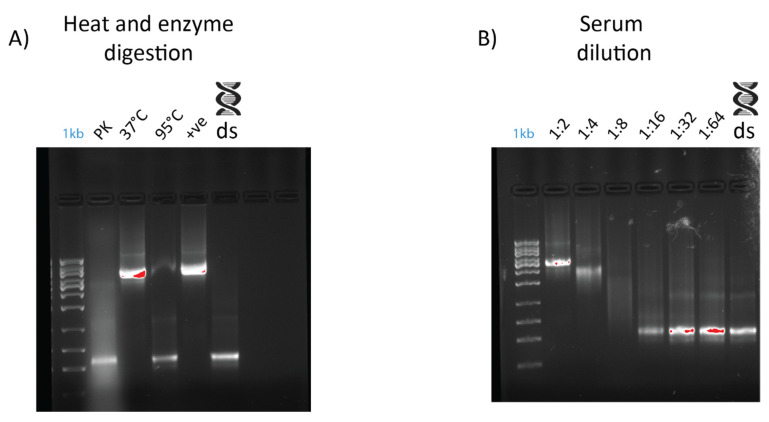
Effects of dilution, heat treatment, and enzymatic digestion on dsRNA–protein gel mobility shift. Serum was extracted from *Cherax quadricarinatus* and incubated with 400 ng of dsRNA for gel shift assays. (**A**) Serum was diluted to 1:2, 1:4, 1:8, 1:16, 1:32, and 1:64 ratios and incubated with the same quantity of dsRNA with a negative control of dsRNA + water as a comparison. (**B**) Serum was subjected to digestion by proteinase K overnight at 37 °C (“PK” lane), with a control at 37 °C overnight (“37 °C” lane) to ensure any serum degradation was a result of enzyme activity rather than temperature. Serum was also heated to 95 °C (“95 °C” lane) for five minutes to test the stability of dsRNA binding components and determine if they were proteinaceous in nature. A positive control (“+ve” lane) of fresh serum and dsRNA and a negative control of dsRNA and water (“ds” lane) were included as a comparison.

**Figure 4 ijms-21-06967-f004:**
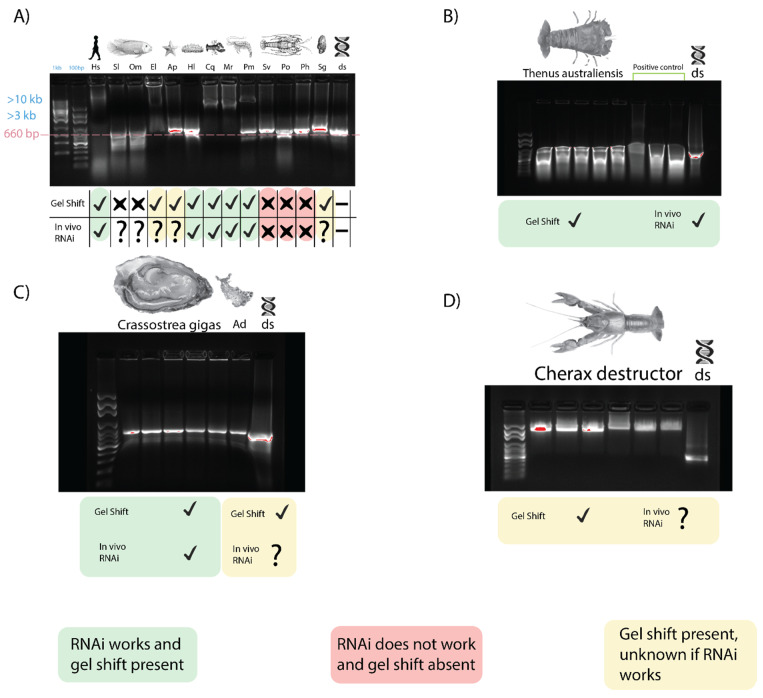
Gel shift trend across phyla. For all lanes, sera (with appropriate anticoagulants when needed) were isolated and incubated with dsRNA for 10 min before being subjected to gel electrophoresis to identify dsRNA binding capacity. A green mark implies a gel shift was observed, and dsRNA-induced exogenous RNAi is functional. Red implies RNAi has not been achievable despite repeated attempts and no gel shift was seen. Yellow implies a gel shift was seen, but data on RNAi capacity is lacking. (**A**) The dsRNA binding activity of sera from invertebrate and vertebrate species. From left to right are *Homo sapiens* (Hs)*, Seriola lalandii* (Sl)*, Oreochromis mosambicus* (Om)*, Epinephelus lanceolatus* (El)*, Acanthaster planci* (Ap)*, Holothura leucospilota* (Hl)*, Cherax quadricarinatus* (Cq)*, Macrobrachium rosenbergii* (Mr)*, Penaeus monodon* (Pm)*, Sagmariasus verreauxi* (Sv)*, Panulirus orantus* (Po)*, Panulirus homarus* (Ph)*, Saccostrea glomerate* (Sg), and a dsRNA control. (**B**) Serum from *Thenus australiensis* and dsRNA with a positive control from *Bombyx mori*, where this phenomenon has been documented previously. (**C**) Serum dsRNA binding of *Crassostrea gigas* and *Aplysia dactylomela* (Ad). Gel shift is modest, but present in both mollusks. (**D**) Serum from *Cherax destructor* and dsRNA.

**Figure 5 ijms-21-06967-f005:**
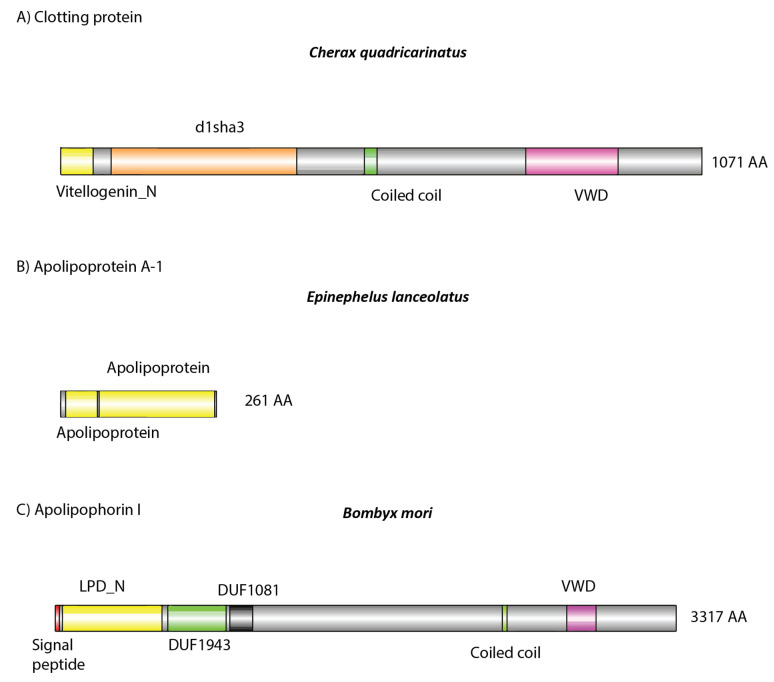
Domain architecture of detected putative (**A**,**B**) and known (**C**) dsRNA binding proteins: (**A**) identified in *Cherax quadricarinatus*, (**B**) in *Ephinephelus lanceolatus*, and (**C**) in *Bombyx mori*. (**A**,**B**) Amino acid sequences were retrieved from our mass spectrometry results. (**C**) Retrieved from publicly available sources (https://www.uniprot.org/uniprot/G1UIS8). (**A**) The protein was identified as a clotting protein, which possesses a lipid binding domain (yellow), as well as a coiled coil and von Willebrand factor type D (VWD) structure, which may facilitate systemic dsRNA transport. This same structure was observed in Apolipophorin 1, which is known to bind dsRNA [33,35]. In *E. lanceolatus*, Apolipoprotein A-1 was identified, which in humans can bind nucleic acids [34]. It also possesses lipid binding domains (yellow), which may also be involved in dsRNA binding when complexed with other serum proteins.

**Figure 6 ijms-21-06967-f006:**
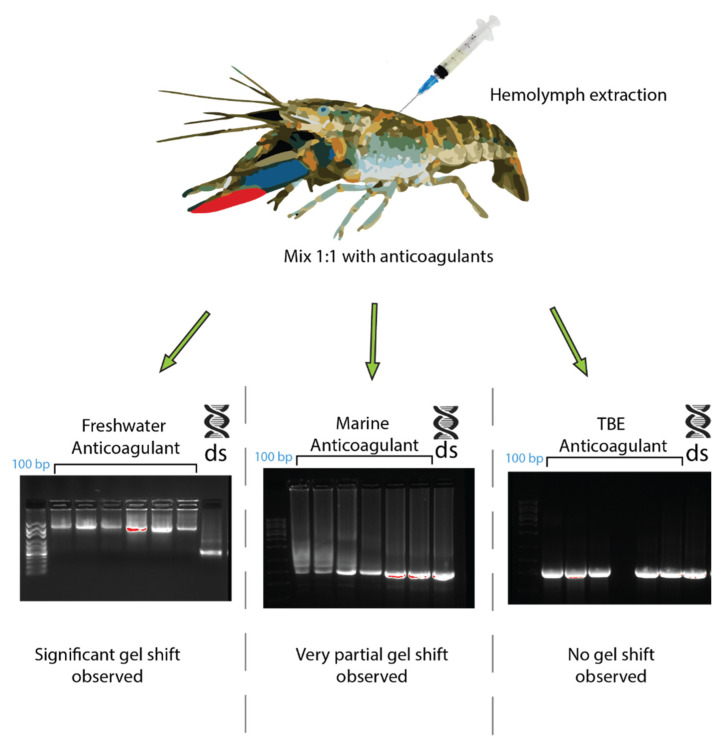
The effects of different anticoagulants on dsRNA–protein complex formation in *Cherax quadricarinatus*. The hemolymph from *C. quadricarinatus* was extracted and mixed with 1:1 volume of either freshwater, marine buffer, or TBE buffer anticoagulant on ice. The hemolymph was then centrifuged and serum was collected and incubated with 400 ng of dsRNA. After 10 min of incubation, samples were run on 1.2% agarose and the difference in binding capacity was observed. In each gel image, the first lane represents a 100 bp ladder, the next seven lanes represent dsRNA + serum with the respective anticoagulant, and the final lane is a dsRNA + water control.

**Figure 7 ijms-21-06967-f007:**
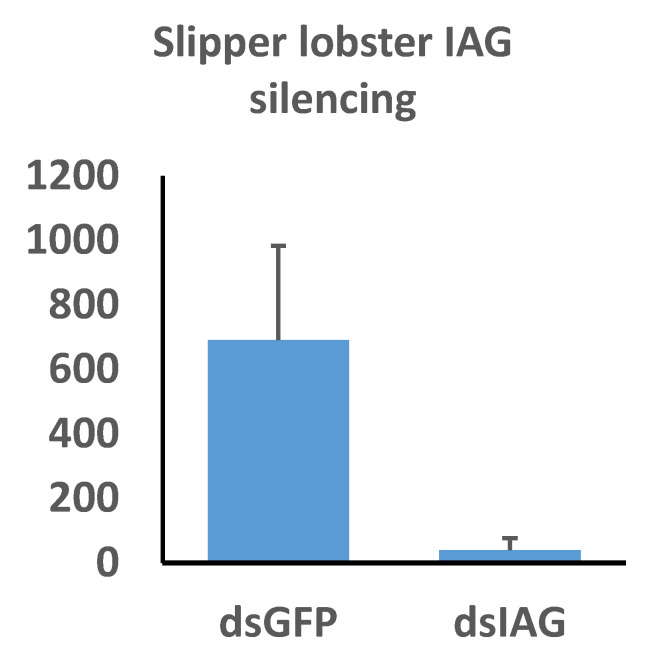
The effects of insulin androgenic gland hormone (IAG) silencing on IAG expression in *Thenus australiensis*. Expression relative to 18s is displayed on the y-axis, while dsRNA treatments are displayed on the x-axis. Juvenile male slipper lobsters were injected once with GFP dsRNA or Taus-IAG dsRNA. Then, 24 h later, the 5th walking legs of the animals were dissected and RNA was extracted and reverse-transcribed to cDNA. Following a qPCR assay, expression relative to 18 s showed significant IAG knockdown in the group injected with Taus-IAG dsRNA (~95%), indicative of robust and highly effective gene silencing.

**Figure 8 ijms-21-06967-f008:**
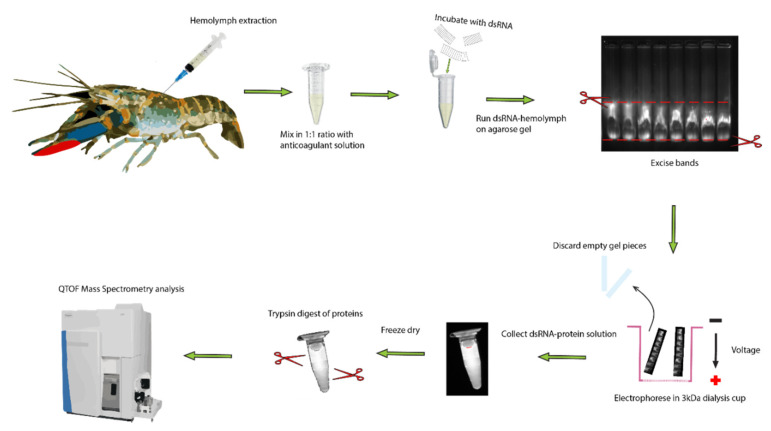
Extraction and identified of dsRNA binding proteins in serum of crayfish. The hemolymph was drawn from the ventral abdominal sinus of a crayfish and mixed with 1:1 volume of freshwater anticoagulant on ice. The hemolymph was spun down at 5000× *g* for 5 min to remove hemocytes and cell debris. Then, 200 µL of serum was incubated with 20 µg of dsRNA and run on 0.8% agarose for 50 min at 100 V. The band was then excised under UV light and placed in a 3 kDa dialysis cup, which was placed back in the gel running tank and electrophoresed for another 50 min at 100 V. The resultant solution containing dsRNA bound to protein was collected and lyophilized. Following this, the proteins were trypsinized and analyzed via mass spectrometry.

**Table 1 ijms-21-06967-t001:** Key proteins identified from mass spectrometry analysis. All proteins listed possess lipid binding or dsRNA binding capacity based on literature and domain analysis. Prior research has implicated lipid binding proteins in nucleic acid binding, so any identified proteins with this function or dsRNA binding potential were noted. Proteins were identified with NCBI BlastP, and data such as e-values and percent identities were recorded. Numbers of peptides and coverage from the mass spectrometer were noted as well.

Species	Proteins Identified (via BlastP)	No. of Peptides	Sequence Coverage	e-Value	% Identity
*C. quadricarinatus*	clotting protein precursor [*Pacifastacus leniusculus*]	1	1	0	50%
*E. lanceolatus*	apolipoprotein A-Ib [*Epinephelus lanceolatus*]	10	24	0	99.62%
*E. lanceolatus*	14 kDa apolipoprotein, partial [*Epinephelus bruneus*]	7	47	1 × 10^−87^	89.51
*B. mori*	apolipophorin III precursor [*Bombyx mori*]	2	14	7 × 10^−131^	100%
*B. mori*	low molecular 30 kDa lipoprotein PBMHP-6 [*Bombyx mandarina*]	1	4	0	97.66%

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
