# Peer review of "Double-Stranded RNA Binding Proteins in Serum Contribute to Systemic RNAi Across Phyla—Towards Finding the Missing Link in Achelata"

_ijms, 2020, doi:10.3390/ijms21186967_

Round 1

Reviewer 1 Report

Banks et al. described the existence of various RNA binding protein in invertebrate hemolymph and discussed their existence to the feasibility of systemic RNAi techniques in respective organisms. The paper covers quite an interesting topic but I found that the flow of study is not so well arranged to reach the goal of the study. 

Major points:

The presentation of electrophoresis data is not so sophisticated.  The lanes could be numbered and the samples should be well described according to the order of lanes. The DNA marker used was only 1 kb ladder, but not so well separated. The running condition was not so constant, since the 4 panels in Figure 3, markers could not be seen in the same way. 

The statements on retarded bands should be carefully considered. For instance, L.130: a band of >8kb,  L.146 A band of >10kb. The band must be constituted from 660 bp dsRNA bound with certain RNA-binding proteins; so should be described like, "the retarded bands with a mobility slower than 8kb DNA size marker in this condition."

Also, to be strict, the authors should know that dsDNA and dsRNA will show different size-dependency for electrophoretic mobility. DNA size markers could work just for convenient standard for dsRNA. 

L.136-138:  The authors described a very important result but without direct data. They SHOULD show the pre-heated serum did not affect dsRNA electrophoretic mobility. It only could show that  protein material is really involved.

L.182-3: I could not understand a slight inhibition, not complete, the authors might mean.  If that is the case, the authors should have tried to add a dilution series of sera to the sample to see the results.  The more, the slower mobility.

Big issue: The authors described many different aquatic animals even fish and insects, mollusks.  What were the main target organisms for them?  The title says Achelata.  So the whole story did not seem consistent all the way.

Minor points:

Figure 5:  100 bp DNA ladders could not be seen in the 2nd and 3rd panels. Their   appearance gave the readers invisible standards to believe the different electrophoresis were conducted in a similar way.  It is important to make it because the authors are discussing significant, very partial or no gel shifts based on the panels. The same thing goes to Figure 3 as well.

Pagination was not conducted through the whole.

The second L.46: what is Tori-IAG?  No details even in Materials and Methods section.

Figure 7 is not always necessary; it is rather general flow of experiments.

The second L.202, 203, 218, 228: Possibly the authors mentioned the same dsRNA but used quite a different word to describe each time.

The meaning of first L.132-4  was not clear to me.

Second L.43:  "the established trend" is not clear.

Author Response

Response to Reviewer #1 comments

Banks et al. described the existence of various RNA binding protein in invertebrate hemolymph and discussed their existence to the feasibility of systemic RNAi techniques in respective organisms. The paper covers quite an interesting topic but I found that the flow of study is not so well arranged to reach the goal of the study. 

Major points:

The presentation of electrophoresis data is not so sophisticated.  The lanes could be numbered and the samples should be well described according to the order of lanes. The DNA marker used was only 1 kb ladder, but not so well separated. The running condition was not so constant, since the 4 panels in Figure 3, markers could not be seen in the same way. 

We respectfully challenge this point. The lanes are labelled with the species name in full, an abbreviated form or graphics. The figure legend also specifies what every lane on every gel refers to, limiting all possible ambiguity. In our view, adding numbers would clutter the figures, thereby reducing their quality.

The DNA ladder is included only to fit convention, as the dsRNA control on the rightmost lane of every gel is the more important marker. The varying migration of the ladder does not retract from the findings of this study. Constant running conditions were used for all gels, as described in our methods (1.2% for visualisation of gel shift, and 0.8 % for protein extractions, all at 100 V for 50 minutes).

The statements on retarded bands should be carefully considered. For instance, L.130: a band of >8kb, L.146 A band of >10kb. The band must be constituted from 660 bp dsRNA bound with certain RNA-binding proteins; so should be described like, "the retarded bands with a mobility slower than 8kb DNA size marker in this condition."

We have made changes in accordance with this comment. The first L:130, the first L:152, the first L:182, the first L: 202, the second L:19, and the second L:35 have been edited to better suit the language choices advised in this comment.

Also, to be strict, the authors should know that dsDNA and dsRNA will show different size-dependency for electrophoretic mobility. DNA size markers could work just for convenient standard for dsRNA. 

We appreciate and acknowledge this point. While it is true dsRNA and DNA will migrate differently, the more important result is to compare the dsRNA control and the dsRNA+ serum samples. The ladder was included not as an important measurement, but rather because convention dictates.

L.136-138:  The authors described a very important result but without direct data. They SHOULD show the pre-heated serum did not affect dsRNA electrophoretic mobility. It only could show that  protein material is really involved.

L.182-3: I could not understand a slight inhibition, not complete, the authors might mean.  If that is the case, the authors should have tried to add a dilution series of sera to the sample to see the results.  The more, the slower mobility.

We addressed both comments by adding an additional figure (Figure 3) which shows the effects of enzymatic digestion and temperature on the gel mobility shift, as well as the effects of diluting the sera. Specifically with relation to the slight inhibition, we re-worded it to be ‘modest’ (now in line 201). The molluscs show a very modest gel shift, making them a less ideal model to investigate the binding mechanism.

Big issue: The authors described many different aquatic animals even fish and insects, mollusks. What were the main target organisms for them?  The title says Achelata.  So the whole story did not seem consistent all the way.

We show in this study compelling evidence indicative of correlation between gel shift migration and gene silencing efficiency. As stated above, decapods appear to show either very significant gel shift (bands appear at >10kbp, depending on species), or, no gel shift at all, in some other species. This correlation, observed across phyla, is a highlight of this study, yet the focus is on achelata, where closely related species show different capacity for gene silencing, clearly correlated with gel shift. This allowed us to start broad with a clear correlation, then focus on the mechanisms in species which we have better capacity to investigate. The cross phyla data were included for two reasons: 1. It makes the correlation between in-effective gene silencing and lack of a gel shift in spiny lobster species far more striking, as it appears their lineage branched off in terms of antiviral immunity which helps explain this result, and 2. Provides insights into how systemic RNAi works in many different taxa.

Minor points:

Figure 5:  100 bp DNA ladders could not be seen in the 2nd and 3rd panels. Their   appearance gave the readers invisible standards to believe the different electrophoresis were conducted in a similar way.  It is important to make it because the authors are discussing significant, very partial or no gel shifts based on the panels. The same thing goes to Figure 3 as well.

As stated above, the more important marker is the dsRNA control. The ladder was included for convention, and all gels were done with identical conditions. The gel imaging software was optimised to best visualise the bands, not the ladder hence its faint visibility.

Pagination was not conducted through the whole.

Editors have fixed this.

The second L.46: what is Tori-IAG?  No details even in Materials and Methods section.

We have now defined it at the second L:47. Tori referred to Thenus orientalis which is now redefined as ‘Taus’ which refers to Thenus australiensis.

Figure 7 is not always necessary; it is rather general flow of experiments.

Given the nonconventional methodology, we believe it is appropriate to include for clarity.

The second L.202, 203, 218, 228: Possibly the authors mentioned the same dsRNA but used quite a different word to describe each time.

The meaning of first L.132-4  was not clear to me.

We have reworded for better clarity in the first L: 132-134.

Second L.43:  "the established trend" is not clear.

We have reworded for better clarity at the second L:42.

Reviewer 2 Report

This is an interesting study with some technical loopholes. Below are my specific comments.

  1. Different amount of serum should be used to show dose dependent shift of dsRNA mobility in Figure 2.
  2. Why the RNA amount in the dsRNA control lane is so less compared to in Spiny Lobster lanes?
  3. Clearly amount of total dsRNA is not same in different lanes of Figure 3A. Then how mobilityshift can be compared among different species samples.
  4. Why only EMSA? Some of these results should be repeated using RNA  pull down assay.
  5. Where are RNAi experiments and Humoral response data?
  6. Legends are incomplete, eg. Markings and pictures in Figure 3 are not well explained in the Figure 3 legend.                                                                      

Author Response

Response to Reviewer #2 comments

This is an interesting study with some technical loopholes. Below are my specific comments.

  1. Different amount of serum should be used to show dose dependent shift of dsRNA mobility in Figure 2.

We have added a figure showing the effects of serum dilution on the gel shift. See figure 3 on the first line 167.

  1. Why the RNA amount in the dsRNA control lane is so less compared to in Spiny Lobster lanes?

The same amounts of dsRNA have been used which is stated in the methods. The reason it may appear less could be due to numerous confounding factors such as loading buffer not homogenizing with dsRNA, salts in the buffers and samples, etc. We offer assurance that in each gel the exact same quantity of RNA was used.

  1. Clearly amount of total dsRNA is not same in different lanes of Figure 3A. Then how mobilityshift can be compared among different species samples.

See the above comment. We most definitely used the same quantity of total dsRNA, as using different amounts like you said would not offer fair comparison.

  1. Why only EMSA? Some of these results should be repeated using RNA  pull down assay.

EMSA and mass spectrometry are suitable methods for identifying potential dsRNA binding proteins. Following EMSA, we have extracted the bands and performed LC-MS/MS analysis to uncover the proteins involved in RNA binding.

  1. Where are RNAi experiments and Humoral response data?

We have edited Figure 4 from ‘humoral response’ to ‘gel shift’, to avoid confusion. The RNAi experiments were either performed by our group, or have been cited from the literature, as referred to in the text.

  1. Legends are incomplete, eg. Markings and pictures in Figure 3 are not well explained in the Figure 3 legend.

We have added more detail to the figure 3 legend to enhance clarity. Each lane is either named with the species name in full, an abbreviation, and/or a graphic to remove ambiguity and what each lane refers to is clearly explained in the figure legend.

Round 2

Reviewer 1 Report

The authors have edited the ms and improved many points raised by the reviewer.  I respect some novel findings in this work.

Author Response

We thank the reviewer for his valuable comments.

Reviewer 2 Report

The major issue I raised about the loading has not successfully addressed. 

Author Response

Dear Editor, Thank you for this response. I find it very peculiar that this is the decision made by the Editors (Major Revision), given the reviewers' comments.   Reviewer #1 commented: "The authors have edited the ms and improved many points raised by the reviewer.  I respect some novel findings in this work.".   This resonates well with our diligent response to the first round of reviews.   Reviewer #2 commented: "The major issue I raised about the loading has not successfully addressed.".   This is very vague. It is not clear what the reviewer refers to. Given that we provided ample explanation to support our findings in the first round of review, we find this general remark not appropriate.I will appreciate it if you could kindly ask the Editors to reconsider their decision, based on Reviewer #1 indication that this is clearly a novel research and the revised version addressed all feedback well. To the very least, I would kindly and respectfully ask you to please contact Reviewer #2 to provide a detailed explanation of what this comment refers to.   We have clearly specified that the loading of the dsRNA and hemolymph was equilibrated for volume and quantity and have provided an additional figure in the previous review round to support our findings further.   Best regards, Tomer